# Developmental Assets and Career Development in the Educational System: Integrating Awareness of Self-Identity, Knowledge of the World of Work and the SDGs in School Programs

**DOI:** 10.3390/bs14020109

**Published:** 2024-02-01

**Authors:** Teresa Maria Sgaramella, Lea Ferrari

**Affiliations:** Department of Philosophy, Sociology, Education and Applied Psychology, University of Padova, Via Venezia 14, 35137 Padova, Italy; lea.ferrari@unipd.it

**Keywords:** self and social systems, SDGs, future identities, school-based interventions, adolescents

## Abstract

Individuals are embedded within systems that possess contextual or ecological developmental assets. Psychosocial assets refer to beliefs that enable positive responses to challenging situations and growth despite adversity, such as hope and a future orientation towards positive attitudes and expectations, as well as persistence and the ability to thrive. Career-related assets refer to career-related resources that characterize career decision-making processes and the world of work, such as the ability to negotiate transitions successfully as well as to tolerate and cope with uncertainty by increasing one’s flexibility and autonomy. This study investigated the effectiveness of two sets of psychoeducational activities in promoting positive attitudes and resources, developmental assets that are useful to strengthen students’ personal resources and shaping their future. This study also highlighted sensitivity to change in personal and career-related developmental assets. Using a mixed design approach, 108 students with an average age of 13.91 years were asked to participate in two psychoeducational activities. The first activity focused on developing a positive future self-identity and the second activity on career exploration and knowledge about the world of work. The results show that each of these two activities support the development of psychological assets as well as of a perspective that addresses complex dynamics and that may reduce inequalities.

## 1. Introduction

Climate change, migration, poverty, and digital transformation are considered global trends that require societies to adapt at a speed never seen before. Regarding digital transformation, artificial intelligence (AI) is now considered a driving force for the economy and society, impacting about 40% of workers globally and 60% in advanced economies [1], but as AI negatively affects half of workers in advanced economies, a polarization effect is expected. This negative effect is higher for older workers and people with low levels of education. The COVID-19 pandemic intensified existing concerns and revealed itself as an additional challenge, thereby accelerating the onset of a brittle, anxious, nonlinear, and incomprehensible (BANI) world. This new concept of a BANI world, proposed by Cascio [2], denotes the global chaos activated by the pandemic in all life spheres, including the impossibility of identifying unexpected situations as risks, the rise in dysfunctional functioning exacerbated by instability, the high heterogeneity of cause-and-effect relations, and the difficulties in convenient decision-making processes and systematic analysis. The emerging related literature shows that the features of this new model should be considered in several life domains and health sectors [3,4,5]. Accordingly, education should develop the knowledge, skills, and attitudes needed to effectively cope with these complexities and should adopt a holistic approach that allows learners to increase both their contextual knowledge and self-knowledge, as well as their interconnections [6]. This education should result in the fruitful improvement in the developmental assets—that is, in the development of a set of skills, competencies, and values that are interconnected and that can positively frame the future lives of children and youths [7,8].

Focusing on the field of career education, diachronic and systemic perspectives are helpful in addressing the actual world of work complexities and in understanding how youth affords meaning both to their experiences in the several systems that they live in and to their career construction processes [9,10]. These perspectives could also support the development of skills that enable them to handle sudden changes in direction and look to the future with hope. Alongside the development of positive attitudes towards and psychological resources for the future, attention should be paid to the cultivation of knowledge of the context and of the world of work. Howard and Ferrari [11] recently emphasized this need to simultaneously consider dimensions associated with both personal and career aspects. Their proposal, which combines social and emotional learning with career knowledge and development, assumes a preventive focus supporting youths in gating into the future with an integrated career and life identity.

Based on these integrated perspectives, in this study, we investigated if and to what extent two paths of activities—one focused on a positive future self-identity development and the other on career exploration and knowledge about the world of work—impact developmental assets. We also examined the sensitivity of these assets to change. Since the pedagogical sequence of learnings related to these two paths of activities was accompanied by psychological exercises designed to lead to a higher self-awareness and personal development, these activities, which we propose under the umbrella of career education, could be considered psychoeducational [12]. We highlight in this paper the specific and diverse impacts of these two career education paths and how they contribute to building steppingstones for an integrated and positive future self.

### 1.1. Self-Identity and Living Contexts

From childhood to adolescence, individuals’ experiences are grounded in the construction of self-identity. According to the perspectives of constructivist and developmental systems [13], to support children and youths in constructing their self-identity, we need to adopt an integrated perspective that considers all aspects of their development and the mediating role of their many relevant contexts, including their family status at birth and their resulting opportunities. Thus, understanding the antecedents and outcomes of the process of the dynamic interactive engagement of youths with their environments is crucial.

Individuals are embedded within proximal to distal systems or contexts that, according to Bronfenbrenner [14], act conjunctively to either support or detract from the individual’s environment and outcomes. Each of these systems possesses assets, termed contextual or ecological developmental assets. These assets comprise other individuals, the social system, institutions, and opportunities for interpersonal interaction and collaboration [15]. Relationships and contexts seem to shape learning and development, thereby contributing to self-identity development [16]. On the other hand, awareness of the self, others, and the environment is pivotal to developing the full range of human knowledge and actions necessary for individuals to face a complex and unpredictable future [17] and to understand and negotiate personal opportunities and obstacles as they grow up [18]. Thus, as suggested in the literature [19], awareness of the self and others is a key component in navigating the challenges to positive development.

Children can already identify influences in their living contexts. Involving them in a reflective activity on diverse systems of influence and asking them to identify, represent, and organize the most important influences for them can first highlight the influences that they intentionally recognize and select as relevant for them, both related to themselves and social systems. They can also actively use them in constructing their system of influences. The potential impact of these influences on children’s intentional career development learning has recently been analyzed, highlighting the role of challenges, such as the COVID-19 pandemic, on their attention to the self and meaningful adults, active engagement in social activities, and openness to the world of work [10].

Therefore, psychoeducational activities aimed at promoting the development of a positive identity among children might be specifically designed to support the development of awareness of the self and of the relationships between the self and individuals in the meaningful systems around the children. Additionally, following a lifelong learning approach, the focus of such psychoeducational activities should be future-oriented, informing goals and aspirations, as well as the different versions of the self in various hypothetical futures [20,21].

### 1.2. Career Exploration and Knowledge about the World of Work

Several diverse frameworks have been proposed to help scholars to develop a career education that addresses the complexities of our BANI world. For this study, we chose to use the lifelong learning framework, which invites participants to reflect both individually and collectively on the past, present, and future [22]. More specifically, and following the lines of action that Carosin and colleagues [22] proposed under the goal-oriented approach to learning, career education in the school context should promote knowledge and skills in three main areas: the self and others, the educational and professional worlds, and, borrowing from recent approaches to career counselling [15], systems of influence. Career exploration and occupational knowledge should be explored not only through learning experiences that address the goals of decent work, social justice, and sustainability [23], but also considering the Sustainable Development Goals (SDGs). The adoption of a lifelong perspective offers a space for an in-depth understanding of the role of human work activities, as well as for expanding the way people relate to the world as a house for all living things, with the aim of both individually and collectively empowering people and making them active agents of social change [22].

Career exploration is conceptualized today as a lifespan process that starts at an early age and enables children and adolescents to build their reasoning and knowledge about the world of work as well as about the self. Pertaining to the world of work, having an in-depth understanding of it, including of its challenges and changes, is important in developing career interests, finding job opportunities, and building a career identity [24,25]. Competencies that focus on the role of learning and work in a person’s life are being paid increasing attention, as seen in documents such as the Australian Blueprint of Career Development [26], where the Learning and Work Exploration competencies include understanding that occupational learning occurs across one’s life and contributes to one’s future work by exploring a range of career information. Research has shown that career exploration grows and becomes more accurate with age [27,28] and supports the achievement of an accurate understanding of occupations, as well as youth engagement in informed decision-making processes. Thus, career exploration is a long process that is continuously challenged by changes in the world of work and requires an investment in education [29].

Although occupational knowledge is quite sparse in the literature, it has been paid specific attention ‘as a bedrock, like the letters of the alphabet, from which more complex and sophisticated knowledge of the world of work and career will emerge’ [30]. Addressing occupational knowledge includes answering questions, such as on the characteristics of different jobs and professions, their educational and other formal qualifications, and ways to access specific positions. It could also require talking to experts in a specific field and checking for available information on websites. In this regard, specific support and guidelines come from the longstanding tradition of job analysis, which helps in systematically acquiring information on occupations despite changes over time in work and working contexts [31,32].

### 1.3. Psychosocial versus Career-Related Assets in Career Development

According to the literature, several personal developmental assets can be identified [7,8]. These assets act as guiding principles for making healthy life decisions. They include positive values, such as caring, social justice, integrity, honesty, and responsibility. They also promote a sense of control and purpose, as well as a recognition of one’s own strengths and potentials, including personal power, self-esteem, and a positive outlook (i.e., a positive self-identity).

Research has shown that children and youths with higher developmental assets have strengths, quality relationships, and interrelated experiences that enhance their sense of self, achievements, and well-being [33]. The outcomes of developmental assets include engagement in career decision-making processes and career sustainability [34]. The first developmental asset, psychological capital (i.e., a construct composed of confidence, resilience, hope, and optimism), deals with resources that support the development of a confident and positive attitude towards challenges in the near or distant future. Its meaningful dimensions are a positive orientation towards the future self, which refers to the tendency to focus one’s mind on the future and on future-valued outcomes [35]; hope, a cognitive process based on the perceived determination to use the available means and self-efficacy to achieve future goals [36,37]; persistence in intentional actions to achieve significant goals with courage, thus sustaining the desire to do things [38,39]; and the ability to thrive in the face of expected adversity, that is, having a resilient attitude driven by belief in the possibility of bouncing back or overcoming some form of adversity and, thus, achieving positive outcomes despite an adverse event or situation [40].

While hope and future orientation capture positive attitudes and expectations, persistence and the ability to thrive explicitly refer to beliefs that are relevant to positively responding to challenging situations and growing despite adversity. Although these two sets of assets are not systematically addressed together, both have been shown to contribute positively to career development learning. For instance, hope is associated with career-related beliefs [41] and educational achievements [42]; the time perspective is considered a prime variable in influencing individuals to proactively set goals and expectations, in regulating their behaviors, and in continually monitoring their performance of given tasks [43,44], which support decision-making processes and favor the future self by prioritizing the future over the present [45]. Persistence has been addressed in the literature on courage [39]. It is related to psychological capital and influences domain-specific and global life satisfaction, flourishing, and subjective well-being [46]. The ability to thrive, referred to as ‘resilience’ in several studies, plays a role in career-related outcomes, suggesting that it has positive influences on the locus of control, purpose in life, self-esteem, interpersonal relationships, and job satisfaction [47]. Adolescents with higher levels of resilience are more motivated to achieve their goals and to plan and implement different solutions, in addition to having more coping skills to achieve their goals [48,49].

The second group of assets is more strictly related to career development and refers to career-related resources that individuals develop by interacting with their environment. These assets are related to the ability of individuals to negotiate transitions successfully, as well as to tolerate and cope with the uncertainties that characterize career decision-making processes and the world of work, by increasing their flexibility and autonomy [50]. Accordingly, the more adolescents think about their future, autoregulate themselves, and manifest a willingness to know more, to persist and to work in a team, the more they demonstrate career adaptability.

Higher levels of career adaptability are associated with several positive outcomes that testify to the crucial role of career adaptability in supporting youths in planning their future and in effective career decision-making processes (e.g., [51,52]). Career adaptability is related to the dimensions of well-being, including life satisfaction, self-esteem, and positive affect (e.g., [51,53]). For example, in a longitudinal study on adolescents, Liang et al. [54] found that higher levels of career adaptability are associated with lower levels of perceived stress over time. Studies have also shown that many individual and contextual factors determine career adaptability [51]. Specifically, in adolescents, it covaries with individual factors, such as self-efficacy [55], educational experiences, and career exploration activities, including job shadowing and informational interviews [56], as well as with contextual factors, such as parental support and economic conditions [57]. As Chen et al. [58] confirmed, both the family socioeconomic status and parenting styles have positive impacts on the development of career adaptability. Additionally, social support comes from one’s family, school, and peers, as well as from larger systems and their policies and measures.

Several psychoeducational programs have been developed across the world to promote career adaptability resources in adolescents (see Table 1). The career construction theory emerged as the theoretical point of reference used for framing group activities and emphasizing narratability and reflexivity to allow participants to understand how their past experiences affected their present and future decisions, and how, by narrating their story, they can gain insights into their progress and identify new goals. The participants were thus involved in many self-assessments, discussions, and reflective exercises. Videos and guide booklets were typically used. The programs varied in the length and number of sessions, which usually ranged from five to seven. The contents were mainly in two main categories: knowledge about the self, such as interests, aspirations, and personal resources; and knowledge about the context, namely barriers and support.

Based on the results of recent studies, knowledge about the self, the world, and society, as well as future orientation and identity in the future world of work, are relevant dimensions to address in career education practices to foster a positive vision of the future and to promote career construction processes.

### 1.4. Purpose of the Present Study

Psychosocial and career-related developmental assets are crucial to promoting positive attitudes and resources that can help young people to strengthen their personal identities and shape their futures.

Thus, the following research questions guided our study: Do psychoeducational activities based on knowledge about the self and living contexts, future representation, or the construction of future goals and knowledge exploration of the world of work impact adolescents’ developmental assets? Do these assets show a differential sensitivity to change? Finally, is the activation of change specific for the two paths proposed? We expect the scores for psychosocial resources (i.e., resilience, future time perspective, and courage) to increase significantly among the participants enrolled in the first curriculum. Similarly, we expect the scores for career-related resources (i.e., dimensions of career adaptability) to increase after the students’ participation in the second curriculum.

## 2. Materials and Methods

### 2.1. Study Participants

The research team proposed this study to the schools with which they were in contact. Two middle and one high schools in Northeast Italy showed interest and were involved in this study. They are in a medium-sized provincial city with various small industries, and they are culturally active. The 108 students who participated in this study were in the second or third year of lower secondary school or the first year of high school. They were divided into two groups and assigned to the two curricula. Table 2 shows the basic characteristics of the participants.

No differences were recorded between the two groups for gender and age (*t* (106) = −0.750, n.s.; and *t* (106) = 0.731, n.s., respectively).

The convenience sample was characterized by the heterogeneity of students attending Italian mainstream education programs and included students with diverse sociocultural backgrounds and school achievements, as well as some students with disabilities and other vulnerabilities. 

### 2.2. Study Design and Procedure

The activities were proposed in the school context under the category of extracurricular educational activities. Schools in the Italian educational system do not hold regular career education activities for students at this age level. Thus, this study could contribute to stimulating attention to career education, starting from the school context in which this study was implemented.

For this study, a mixed measures design was employed. More specifically, the within-participants factor was time, that is, at the baseline versus post-intervention, and the between-subjects variable was the curriculum. Comparing two conditions may provide information and support the drawing of conclusions on the effectiveness of the two paths, as well as reveal whether the observed effects were due to the specific themes addressed or to more general and common dimensions.

In the preliminary session, the participants were informed about the general aim of this study and about the schedule of the activities, which were proposed during school time. Then, the participants were asked to complete the research protocol. One week after the preliminary session, we started this study.

After the school received the written consent from the children’s parents for their children to participate in this study, the participants were randomly assigned to one of the two curricula. Quantitative pre- and post-tests were performed to test the target changes.

All the activities were conducted in line with the current data protection laws on the processing and communication of personal data, namely D. Lgs 196/2003 (Legislative Decree 196/2003, known as the "Personal Data Protection Code) and the EU GDPR 679/2016 (European General Data Protection Regulation 2016/679)**.**

Teachers participated in the activities and contributed to the active involvement of the adolescents. To ensure the confidentiality and anonymity of data, only the two counsellors conducting the activities were provided access to the information needed to match the profiles that emerged from the pre- and post-tests. Additionally, the teachers were provided the results of all the participants.

### 2.3. The Two Curricula

The authors designed the two curricula and validated them through this study. The two curricula had similar structures. Both consisted of four sessions, with one session conducted for 2 h per week. Four main steps were followed in the planning and implementation of the curricula. In the first step, *Goal exploration and knowledge sharing*, the topic was introduced in the first session. In the second step, *Active engagement*, the circumstances in which the behavior or attitude was expected or expressed were described, the actions and tasks that the participants were expected to carry out were defined, and indicators of the achievement of the goals and learning were identified. In the third step, *Reflection time*, the participants were allowed to stop and ponder on the meaning of the activity carried out and its value for them. For instance, to connect the activities to future goals, the participants were asked to think about the implications of the activities for their future work activities. Finally, in the last step, *Share with classmates* and *with family*, the participants, together with the teacher and the conductor, were asked, at the end of each session, to identify ways to impart the ideas from the session with their classmates and with their family to stimulate the generalization of learning. They were typically asked to search for images or words that could best represent some ideas that came out at school and to share them with their parents. Various strategies were used, from simple instructions to examples to ensure active participation.

The first curriculum, *Me and My Systems of Thrust* (in short, *Systems of Thrust*), focuses on developing awareness of and encouraging reflection on sources of influences on the participants that they identified and that were related to the specific system addressed. The theoretical reference was the systems theory framework (STF), a comprehensive metatheoretical framework of career development that emphasizes both content and process—in this study, the dynamic nature of relationships between the individual and the diverse systems they live in, from the closest one, such as individual and social systems, to larger and more complex environmental and societal systems, from the past to the future. The participants created their own stories using *My Systems of Career Influences* (MSCI Adolescent version) [64], a qualitative career assessment instrument based on STF that is proposed for adolescents in group career counselling activities [65]. The students, working through a booklet page by page where maps were proposed, were asked to write down using a word or a sentence the influences they could identify as related to the specific system represented and to mark those they considered as the most important ones.

In the first meeting, titled *Meanings in My Life*, the participants engaged in an activity aimed at discovering themes that were personally relevant to them. The activity consisted of a word association task using a list of words related to work, study, and leisure time. After the activity was explained to the participants, they were asked to associate a target word (e.g., work) with an adjective that they believed best described the word. They shared the associations produced, guided by some questions. During the reflection time, they were helped to think about their own associations and the relationship of those with their lives.

In the second meeting, titled *Me*, *My Story*, and *My Closest Systems*, the participants, after sharing knowledge on the keywords associated with the target themes more frequently provided in the instructions (e.g., interests and strengths), were asked to reflect on themselves, paying attention to their strengths, their uniqueness, and the influences on them of people in their closest environments, such as their family and friends. The participants drew representations of their identified influences on a map and marked the influences that had a specific relevance. During the reflection time, they were asked to comment on what they drew on the map, thus supporting the development of their awareness of their uniqueness and the persons around them and those persons’ influences on them.

In the third meeting, titled *Me and the Larger Systems around Me*, after the participants shared their knowledge of the definitions of the target concepts that were more frequently provided in the instructions, they were asked to draw on the map elements and influences in their lives that came from the people they met and from the situations that occurred in their wider contexts, from the school to the neighborhoods and from the city to the larger communities. Again, during the reflection time, the participants were asked to comment on what they saw on the map, thus supporting the development of an awareness of their personal uniqueness and of the persons around them and those persons’ influences on them.

In the fourth meeting, *My System of Thrusts*, the participants, following an example of the systems of influence developed by a peer, were instructed to return to the single map they had worked on in the previous sessions and to select influences from among those they identified. Then, they were required to develop their own representation of their chosen influences in the *My Systems of Thrust*. Afterward, during the reflection time, they were asked to reflect on the insights they gained by answering a set of reflective questions that enabled them to elicit meaning and learning, alongside actions that they foresaw in their future to keep active and leverage their positive thrusts.

The second curriculum, *Me*, *Work*, and *Our Sustainable Future* (in short, *Sustainable Future*), focuses on the meaning and content of professional activities in a person’s current life context, the person’s knowledge of the SDGs, and the relationship of both to the person’s future as a citizen and a worker who pays attention to decent work and sustainability and who can act responsibly under the guiding principles of social justice [23]. The theoretical reference points in the development of this curriculum were the psychology of working theory and the decent and meaningful work concepts [66] (Blustein and Duffy, 2020). Also considered as a model for implementing sustainability were recent advancements in job analysis [32].

In the first meeting, *Work*, *Study*, and *Leisure Time*, the participants were provided some basic ideas on how work has changed throughout history, after which they were invited to list their ideas regarding the concepts of work and to define what a good job and a good work environment were. For this task, they were provided information on people’s ideas about work. Then, they were asked to focus on their current diverse roles by focusing on their study and leisure-time activities, from which they were asked to reflect on the diverse ways to look at work, study, and leisure time and to describe their own views.

In the second meeting, titled *Knowing People at Work*, the focus was on activities associated with some professions that were proposed for analysis. The participants were asked to describe the activities by indicating the tools used by each professional, who could use the professional’s service, the tasks, and roles that could be associated with the specific job, the places where the professionals implement the activities, and the people that the professional could meet. Finally, the participants were asked to reflect on these dimensions of the work activities analyzed.

The third meeting, titled *The Agenda and Its Goals*, focused on exploring the goals proposed by the 2030 Agenda. The activities required answering questions about the different appropriate actions that facilitate the achievement of the proposed goals. An example of an activity is titled *Find the right key to open the door of ….* Several keys represented specific goals in the Agenda (e.g., health and well-being, gender equality, clean water, green energy, dignified work, and reduction in inequalities), while the doors represented the triple bottom line (people, planet, and profit, with the latter being represented as economic well-being and as an inherent result of providing value to people and to the planet). Finally, the participants were asked to reflect on the diverse actions that could be performed to support the achievement of the goals.

In the fourth meeting, titled *Me and Work at the Time of the Agenda*, after the 17 goals were discussed more extensively, the participants were asked to identify professionals who, through their work, contributed or could contribute to the achievement of each of the 17 goals. An example of a related activity is *My Footprints to the Future*, in which the participants were asked to identify paths that connected a goal of the Agenda to actions (footprints) that a professional could carry out to achieve the goal. Then, the participants were invited to reflect on the activities in which they could see themselves involved in the future and to color their footprints.

### 2.4. Measures

Two sets of quantitative tools, developed or adapted, and validated with adolescents in the Italian context, were used to identify changes and learning. In line with the aims of this study, the first set referred to the dimensions of psychosocial assets and the second set to career-related assets.

For the psychosocial assets, using the Design My Future questionnaire [67], two assets were addressed: future time orientation (e.g., *Looking ahead and thinking about what will happen in the future makes me feel full of energy*) and the ability to thrive (e.g., *I think I’m able to challenge the difficult situations that may arise in the future for me*). The participants responded to each item on a scale ranging from 1 (*not strong*) to 5 (*strongest*). The Cronbach’s alpha was 0.82.

Hope was assessed using a subscale from the Vision About the Future questionnaire [68]. The participants responded to each item (e.g., *In the future*, *I will be involved in very important projects*) on a scale ranging from 1 (*not strong*) to 5 (*strongest*). The Cronbach’s alpha was 0.81.

Persistence was addressed using the Norton and Weiss [39] scale in the Italian adaptation [69]. It consists of 13 items. The participants indicated the degree of their agreement with each item on a scale ranging from 1 (*strongly disagree*) to 7 (*strongly agree*). Examples of the items are *I act courageously* and *Even if something scares me*, *I will not back down*. The Cronbach’s alpha was 0.77.

Career-related assets were assessed using the Career Adapt Abilities Scale—Italian Form [70]. The individual scores of the participants who responded to each item on a Likert scale ranging from 1 (*not strong*) to 5 (*strongest*) were combined into a total score that indicated career adaptability, and the items were grouped into four six-item subscales that measured the following adaptability resources: concern (e.g., *Realizing that today’s choices shape my future*), control (e.g., *Counting on myself*), curiosity (e.g., *Investigating options before making a choice*), and confidence (e.g., *Working up to my ability*). The Cronbach’s alpha for the four subscales ranged from 0.73 to 0.88.

### 2.5. Analyses

The preliminary analyses showed differences between the two groups, which were recorded at the baseline using the following psychosocial resources: hope (*t* (106) = −0.750, n.s.), ability to thrive (*t* (106) = −0.750, n.s.), and persistence (*t* (106) = −0.750, n.s.). The *t*-test showed similar patterns for the level of concern (*t* (106) = −0.998, n.s.), control (*t* (106) = −0.496, n.s.), curiosity (*t* (106) = −1.279, n.s.), and confidence (*t* (106) = −1.257, n.s.).

No deviation from the linearity was found in the initial test performed with the scatter plots. The skewness and kurtosis values were checked for normality and the homogeneity of the variance–covariance matrices, and no statistically significant violation was found.

For each participant, changes in the scores from the baseline to the end of the educational activities were then analyzed. To examine the effect of the curriculum, we carried out a repeated-measures mixed Analysis of Variance with the type of curriculum as the between-participants factor and time (pre vs. post) as the within-subjects variable. The analysis was conducted to determine the general effect of psychoeducational activities on developmental assets and the effect of the type of psychoeducational activities on the psychosocial and career-related resources. We used SPSS version 28 (IBM, Armonk, NY, USA).

## 3. Results

A multivariate analysis was conducted to compare the magnitudes of the changes in the scores for the developmental assets (i.e., the psychosocial and adaptability resources) reported by the participants in the *Systems of Thrust* group and the participants in the *Sustainable Future* group. Table 3 shows the main results of the psychosocial assets.

Regarding the psychosocial assets, a significant multivariate effect was found for time (*F* = 6.632, df = 4.103, *p* < 0.001, *η^2^* = 0.205), which showed that participation in the activities had an effect, but the type of activities (*F* = 1.117, df = 4.103, n.s.) and the interaction between the curriculum and the time had no effect (*F* = 1.175, df = 4.103, n.s.). Indeed, the follow-up ANOVAs revealed significant changes after participation in the activities, as shown by the increase in the future time perspective (*F* = 17.784, df = 1.103; *p* < 0.001, *η*^2^ = 0.144) and persistence (*F* = 11.301, df = 1.103, *p* < 0.001, *η*^2^ = 0.096), but not in hope (*F* = 1.677, df = 1.103, n.s.) and the ability to thrive (*F* = 3.151, df = 1.103, n.s.). The inspection of the means in Table 3 revealed that, for both groups, the scores on the future time perspective and persistence increased, but the increases were larger for students who participated in the activities focused on a *Sustainable Future*.

Regarding the adaptability resources, significant multivariate effects were seen for the main effects of the psychoeducational activities (*F* = 5.582, df = 4.103, *p* < 0.05, *η*^2^ = 0.087) and time (*F* = 3.113, df = 4.103, *p* < 0.018, *η*^2^ = 0.108). The interaction between the two curricula and the time was insignificant (*F* = 1.134, df = 4.103, n.s, *η*^2^ = 0.042). The follow-up ANOVAs revealed that, after participation in the activities, the scores for *concern* (*F* = 6.400, df = 1.103, *p* < 0.013, *η*^2^ = 0.057) and confidence (*F* = 7.47, df = 1.103, *p* < 0.007, *η*^2^ = 0.066) significantly changed, but not the scores for control (*F* = 0.14, df = 1.103, n.s.) and curiosity (*F* = 1.833, df = 1.103, n.s.). The inspection of the means (Table 4) revealed that the scores of the participants in the *Systems of Thrust* group for confidence increased, as did the scores of the participants in the *Sustainable Future* group for their level of concern.

## 4. Discussion

The main goal of the current study was to examine the sensitivity of developmental assets to change as promoted by the proposed career-related psychoeducational activities from the beginning of adolescence. These improvements are considered strengths that significantly contribute to building steppingstones for an integrated and positive future self. Additionally, we compared the specific impacts of the psychoeducational interventions.

The proposed activities consisted of two different school programs, each with four sessions a week. The first path focused on exploring personal strengths and resources using a systemic approach. The reflective process aimed at increasing awareness of self-identity and interconnections with several systems of influence. The second path taught students new ways of exploring the world of work, as well as how to reflect on the meaning of work and leisure time and their contribution to the achievement of the SDGs as agents of change.

Regarding the first question, the results clearly show that the proposed psychoeducational activities impacted the developmental assets targeted in this study. The significant differences between the scores in the pre- and post-tests show an increase in the dimensions addressed. These results signify the value of offering learning experiences in the career education field during adolescence when the knowledge and skills acquired during childhood are consolidated, while some are redesigned, and new ones develop.

The assets also showed a differential sensitivity to change. More specifically, among the psychosocial assets, the tendency to focus on the future and on future-valued outcomes, persistence in intentional actions, and significant goals associated with courage significantly increased at the end of the program. Among the adaptability resources, attention to the future self (i.e., concern) and self-belief in the possibility of achieving career goals (i.e., confidence) emerged as more sensitive to change among the career-related resources. Additionally, as mentioned, this study highlighted a common sensitivity to, and interrelation of these interacting systems with, the level of developmental assets. The results confirm that adolescence is a time window that is useful for supporting youths in projecting into the future and designing their possible paths. Moreover, the results suggest the feasibility of anticipating this type of activities before full adolescence. These types of actions can assume a preventive role by motivating pre-adolescents to devote time and energy to their career development to benefit from it in the future.

As expected, for the last question, the activation of change emerged as specific to the two paths proposed. In the first path, the participants explored influences in the proximal systems (i.e., the personal, familial, and educational systems), alongside the more distal systems as in the Bronfenbrenner model. The results show that this path was effective in increasing the participants’ awareness of themselves and of the influences that led them to their choices and active involvement in life. Moreover, their scores for their confidence in the possibility of achieving their future goals also increased. Overall, these results confirm that activities focusing on systems of influence are effective in supporting the person–context interplay with systems of reference, specifically, with the self, close, and distal systems of reference, in the process of meaning-making. The results also highlight the preventive role of the career education activities proposed, since a better awareness of the self, others, and the context is considered ‘pivotal’ to obtaining the full range of human knowledge and actions [17] needed when individuals face hard times [18] and is critical to navigate challenges to positive development [19].

In the second path, the participants explored the future through the lenses and goals suggested by sustainability principles. The results show that the participants in the *Sustainable Future* program increased their scores for the concern dimension of adaptability, as well as for the future-oriented time perspective and persistence dimensions. These improvements confirm that this path is effective in supporting an active and positive involvement, a future orientation, commitment to preparing for upcoming career tasks or challenges, and a sense of responsibility for actively influencing one’s own development. Following recent general studies in the literature [71], we can speculate that the work conducted on the SDGs helped the participants to develop a new awareness of meaningful action and the possible roles that they may actively play as adults in their future lives and work contexts. Programs based on career-related psychoeducational activities can actively contribute to the achievement of the SDGs and are valued for their role in promoting equality and social justice. Moreover, by increasing psychosocial assets, adolescents also increase their potential to combat challenges to positive development, such as climate anxiety, which is emerging as an increasing source of anxiety and a mental health issue for children and adolescents [72].

In summary, the two activities described in this study succeeded in developing the students’ awareness of, and then the propensity for reflection on, a positive future perspective that addresses complex dynamics and potentially reduces inequalities. Such activities and others like them that direct students’ attention to the systems in which they live and to their possible future identities and professional roles might help them to foresee their potential transformative roles as individuals and as members of their communities [42,72,73].

## 5. Conclusions

As denoted in the title of this paper, career development goals and perspectives in the educational system seems to be moving towards the integration of awareness of self-identity with knowledge of the world of work and the SDGs in programs for adolescents. In this study, both curricula proposed that youths engage in designing career–life paths that conciliate their personal plans with the possibility of a transformation of their environment and living contexts. Both curricula laid the foundations for durable, long-lasting resources and skills that can potentially support the students in facing an uncertain, and eventually, unsupportive work context. Longer and integrated programs could support students in reflecting more and in learning new and effective strategies for adapting themselves to the contexts in which they will live in the future and for becoming active agents of change in such contexts, following the SDGs.

The results provide directions for the integration of formal career education in schools in which marginalized groups of students, in particular, may benefit from career awareness and support. Overall, the activities increased the students’ readiness to consider the complex interrelations that characterize their systems of life and to start including sustainability issues in their career and life projects, in addition to fostering their developmental assets and resources. Thus, schools confirmed as ideal and effective contexts for these actions.

Further studies may investigate more specifically the impact of each psychoeducation activity on the same group of adolescents and analyze the combined impact of the two activities. Although some common effects emerged in this study, the two curricula showed specific and unique effects, and their participants can be expected to demonstrate higher engagement and achievements than students who attend only one program. Finally, the impact of increasing age during adolescence should also be investigated. In several countries, such as Italy, 14-year-old middle school students are required to choose secondary school from several diverse options. Transitioning to a new school setting with new educational paths and curricula may impact the students’ attitudes towards the complex interrelations that characterize their living contexts and large systems of life, as well as their career and life projects.

## Figures and Tables

**Table 1 behavsci-14-00109-t001:** Summary of recent programs promoting career adaptability resources.

Authors	State	Age (years)	Duration (No. of Sessions)	Goal	Methodology
Carvalho et al. [59]	Portugal	12–18	7 (1.5 h each)	Foster a positive vision of the future; knowledge about the self, the world, and society; the career decision-making process; and career planning.	Exercises and guided discussions on career review, interests, and aspirations, designing the Life Career Rainbow and an action plan
Rabie et al. [60]	South Africa	14–15	2 (1st: 2 h; 2nd: 5 h)	Foster self-knowledge, career decision-making skills, contextual awareness, and knowledge about career counselling.	Distribution of a booklet as a guide to the workshop with topics on self-knowledge and context exploration
Santisi et al. [61]	Italy	11–18	6 (time not reported)	Foster self-knowledge and contextual knowledge, optimism, hope, and career adaptability.	Video watching, discussions, and reflective activities
Gülsen et al. [62]	Cyprus	14	5 (1st: 1.25 h; others: 1 h)	Foster key career construction processes, self-exploration, career adaptability, future orientation, a narrative identity, and life satisfaction.	Reflective exercises, discussions, and narration of career stories
Gee et al. [63]	The United States	14–17	64 (50 min each)	Foster interpersonal skills, coping skills, and the skills and knowledge needed to enter a career or post-secondary education.	Team building and self-awareness activities, mindfulness and stress management exercises, goal setting and role model and networking activities, mock interviews, expert panels, a college campus tour, lessons

**Table 2 behavsci-14-00109-t002:** Characteristics of the study participants involved in the two paths.

	Me and My Systems of Thrust	Me, Work, and Our Sustainable Future
Number of participants	59	49
Boys	30 (50.8%)	25 (51.0%)
Girls	29 (49.2)	24 (49.0%)
Mean age	13.98	13.84
(SD)	(1.04)	(1.03)

**Table 3 behavsci-14-00109-t003:** Descriptive statistics for the psychosocial resources by group (i.e., *Me and My Systems of Thrust* and *Me*, *Work*, and *Our Sustainable Future*) and by time of assessment (pre and post).

	Systems of Thrust				Sustainable Future			
	Pre		Post		Pre		Post	
Resource	Mean	SD	Mean	SD	Mean	SD	Mean	SD
Hope	24.73	3.92	24.69	4.06	24.29	3.84	25.53	4.58
Ability to thrive	37.46	6.76	37.69	6.81	37.02	5.86	39.59	5.99
Future time perspective	49.29	8.63	52.53	8.41	47.49	8.54	51.39	9.02
Persistence	68.76	8.89	70.22	9.10	68.71	8.53	73.37	9.21

**Table 4 behavsci-14-00109-t004:** Descriptive statistics for adaptability resources by group (*Me and My Systems of Thrust* and *Me*, *Work*, and *Our Sustainable Future*) and by time of assessment (pre and post).

	Me and My Systems of Thrust				Me, Work, and My Sustainable Future			
	Pre		Post		Pre		Post	
Resource	Mean	SD	Mean	SD	Mean	SD	Mean	SD
Concern	20.71	4.40	20.42	4.02	21.51	3.80	22.47	3.44
Control	22.07	3.02	22.58	3.04	22.39	3.69	22.12	3.81
Curiosity	20.80	4.56	21.56	3.88	21.76	2.83	22.04	3.19
Confidence	21.93	3.42	23.14	4.04	23.67	3.11	24.39	3.75

## Data Availability

The data are unavailable due to privacy restrictions.

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
