# Peer review of "Developmental Assets and Career Development in the Educational System: Integrating Awareness of Self-Identity, Knowledge of the World of Work and the SDGs in School Programs"

_behavsci, 2024, doi:10.3390/bs14020109_

Round 1

Reviewer 1 Report

Comments and Suggestions for Authors

Although this study may contribute to the literature related to students' career development, the study needs to be improved in several ways before acceptance:

1.) You must define "psychoeducational actions" and be consistent with the definition and usage of the term throughout the paper.

2.) Please clearly state your research questions in your introduction, as well as your sample and briefly how you gathered data. Otherwise, the paper is aimless from the introduction.

3.) Please display your participants in a table or matrix so the reader can better understand your study's sample.

4.) You must explain why you chose the questionnaires to administer--are they validated? Were they developed for your population or sample? There must be much more detail here, otherwise your reader will not understand why you chose the assessment and how it answered your research questions.

Comments on the Quality of English Language

English requires moderate editing, as there are numerous subject-verb disagreements and APA 7 errors.

Reviewer 2 Report

Comments and Suggestions for Authors

It is a good paper that highlights relevant issues for child and youth development. The authors prove deep knowledge in the field of Educational Guidance or Counselling at school, and it may significantly contribute to the scientific community with good practices to be replicated.

However, some improvements can be introduced in order to increase the paper's quality:

·        In the Introduction, ax explicit reference to AI and its deep impact could be mentioned.

·        In section 1.1, a reference to Bronfenbrenner’s model would be highly appreciated to support the section’s rationale.

·        In section 1.2, a framework on Educational Guidance and Counselling should be introduced in order to frame the whole paper.

·        In section 1.3, from line 183 to 223, the information could be showed in a table. Besides, the text is too descriptive, some comprehensive information should be added.

·        In sections 2.1 and 2.2, the information is poor. More information about the selection criteria of the school should be added. Besides, more information about the students socio-economic background ought to be  described, since it is a key variable to interpret the results. There is no discussion about the fact that the proposed activities are extracurricular, and the effect of this issue on the whole research. Finally, there is no description on the variables’ control, or a justification on not to consider control groups for both training programs.

·        In section 2.3, there is a lack of explicit information about the training programs: who designed them? Who validated them, and when? Which is the theoretical background behind them? The information could be showed in a table.

·        Both the discussion and conclusion sections seem to be a follow-up of the literature review with very few references to the results 8mainly between lines 441 and 488). There is very little discussion actually. Results should be a clear and prominent part of these sections.

·        Finally, the statement included between lines 518 and 522 could be rephrased, since a 8-hours program in a month can be a significant contribution to the students’ education, but never bring deep transformation of their personality or future expectations in a changing world.

Round 2

Reviewer 1 Report

Comments and Suggestions for Authors

I appreciate your edits and adherence to reviewer feedback, and I feel this manuscript is ready for publication.

Comments on the Quality of English Language

Minor English grammar editing and punctuation needed.